# Hospital homebound students and K-12 online schooling

**Erik W. Black** [1,2] *, **Richard E. Ferdig** [3,4], **April Fleetwood** [5], **Lindsay A. Thompson** [1,6]

**1** Department of Pediatrics, College of Medicine, University of Florida, Gainesville, Florida, United States of America, **2** School of Teaching and Learning, College of Education, University of Florida, Gainesville, Florida, United States of America, **3** School of Teaching, Learning and Curriculum Studies, College of Education, Health, Human Services, Kent State University, Kent, Ohio, United States of America, **4** Research Center for Educational Technology, Kent State University, Kent, Ohio, United States of America, **5** Office of Analysis, Assessment, and Accountability, Florida Virtual School, Orlando, Florida, United States of America, **6** Department of Health Outcomes and Biomedical Informatics, College of Medicine, University of Florida, Gainesville, Florida, United States of America

☯ These authors contributed equally to this work.

* ewblack@ufl.edu

**Data Availability Statement:** Our ability to provide access to the data is limited by Florida law. Family Educational Rights and Privacy Act (FERPA), 20 U. S.C. § 1232g; 34 C.F.R Part 99; and chapter 119 and section 1002.22, Florida Statutes states: Individuals and organizations that receive data

## Abstract

The flexibility afforded by online education may provide opportunities for learners with disability who require absence from traditional learning environments. This study sought to describe how a subset of learners with disability, those with hospital-homebound designation, perform in K-12 online classes, particularly as compared to non-hospital homebound counterparts. A cross-sectional analysis was performed of all Florida Virtual School course enrollments from August 1, 2012 to July 31, 2018. Researchers analyzed 2,534 course enrollments associated with K-12 students who, at the time of their course enrollment, had hospital-homebound designation, and a comparison group of 5,470,591 enrollments from K-12 students without hospital-homebound status. Data analysis showed three important outcomes. First, hospital-homebound designated student academic performance was equivalent to their non-hospital homebound counterparts. Second, however, hospital-homebound course enrollments were 26% more likely to result in a withdrawal prior to grade generation. Third, these withdrawals were potentially mitigated when H/H designated students were enrolled in five or more classes or in classes with five or more students. The results of this study provided evidence that when they can remain enrolled, hospital-homebound learners experience equivalent academic outcomes in online learning environments. These findings suggest that healthcare professionals should be made aware of the potentially equivalent outcomes for their patients. Moreover, virtual schools should seek to identify and create supports for these students.

## Introduction

Given advances in healthcare, more children are surviving illness and disability than in years past [1]. These children represent considerable diversity racially, ethnically, and in the

through the research request process have strict obligations to protect that data and must meet the requirements of both federal and state laws in their usage and handling of that data. The parties receiving this data have separate obligations based on these provisions and must protect the confidentiality and privacy of this information and may not disclose this information to others. The Florida Department of Education requires all requests to comply with the Family Educational Rights and Privacy Act's (FERPA's) research exception (34 CFR Part 99.31[6][i]), which requires that the disclosure of information be limited to organizations conducting studies for, or on behalf of, educational agencies. In addition, the research must meet one of the following three allowable purposes: 1) to improve instruction; 2) to develop, validate or administer predictive tests; or 3) to administer student aid programs. Requests must meet one of these criteria to be approved. In sum, the data is available via public request via Florida's Government-in-the-Sunshine law. This provides the right of access to governmental documents at the state and local level. Data access can be facilitated using the Florida Department of Education External Research Data Request Process: https://www.fldoe.org/accountability/accountability-reporting/external-research-requests/. While this is not standard, we feel that this does not alter our adherence to PLOS ONE policies on sharing data and materials.

**Funding:** The authors received no specific funding for this work.

**Competing interests:** April Fleetwood is employed by Florida Virtual School. No other Florida Virtual School employee played no role in study design, data analysis, or preparation of the manuscript. Florida Virtual School aided in the collection of data and received a draft of the manuscript for review prior to submission. This does not alter our adherence to PLOS ONE policies on sharing data and materials.

heterogeneity of their health-related needs [2]. Promising research suggests that while chronic illness represents a challenge, it does not prevent children from living fulfilling lives. For example, Blackwell et al. examined data from three cohort studies and determined life satisfaction was comparable between those with and without chronic illness. The authors concluded that: "chronic illnesses. . .do not preclude children from leading happy and satisfying lives" [1(p6)].

Nevertheless, growing up with a chronic illness or disability not only requires high levels of medical care, it also places children at increased risk of poor educational outcomes. For example, they are at a higher risk for missed school days, low educational attainment, repeating a grade, and poor mental health outcomes than the typical child [3, 4]. These educational risks also translate to poor occupational outcomes such as difficulties in obtaining or maintaining employment, potentially negatively impacting psychosocial well-being and quality of life [3, 5]. Meeting the educational needs of these children to provide a foundation for their fulfillment and independence requires a robust support system, dedicated resources, and interprofessional collaboration [6].

The dominant paradigm in the United States (US) for educating students with chronic disability and illness encourages inclusion, integrating learners with different needs in the same classroom [7]. However, a small but significant number of students with chronic illnesses or disabilities may not be healthy enough to attend school in a traditional environment. In these cases, hospital/homebound (H/H) instruction may be an appropriate option [8, 9].

H/H instruction is not a new phenomenon; evidence of homebound instructional services for pregnant teens in the US dates to the 1930s [10]. H/H programs provide educational instruction in non-school settings for children who are living with short-term and chronic disabilities H/H programs are publicly funded and supported; this differs significantly from homeschooling where a child's parent or guardian traditionally assumes responsibility for the delivery of educational services. H/H is administered in several different formats, varying between localities and states. Examples include home-based instruction (in which a certified teacher physically visits students in their home) or phone-based instruction. Most recently, following a similar trend in homeschooling, there has been increasing enrollment in one or more online courses [11]. Online instruction can also take many different formats, but most instruction typically involves electronic engagement between a teacher, student, and course materials (e.g., audio, video, text, or other course materials located in a learning management system; see Davis & Ferdig) [12].

H/H environments are considered highly-restrictive; as such, learners wishing to participate in an H/H program must meet qualification standards on a case-by-case basis related to their health impairment [13, 14]. For instance, in the state of Florida, obtaining H/H designation and services requires confirmation from a physician, nurse practitioner, or physician assistant with Florida licensure that the student meets the following criteria:

1. *That the student is expected to be absent from school for at least 15 consecutive school days (or the equivalent on a block schedule) due to a physical or psychiatric condition, or for at least 15 school days (or the equivalent on a block schedule), which need not run consecutively, due to a chronic condition.*

2. *That the student is confined to the home or hospital.*

3. *That the student will be able to participate in and benefit from an instructional program.*

4. *That the student is under medical care for illness or injury, which is acute, catastrophic, or chronic in nature.*

5. *That the student can receive instructional services without endangering the health and safety of the instructor or other students with whom the instructor may come in contact.*

(Exceptional Student Education Eligibility for Students Who Are Homebound or Hospitalized § 6A-6.03020, 2006, p.1).

There is very little published research about H/H programs or the children they serve; this is due, in part, to the relatively small number of learners who receive H/H services during their academic career as well as the varied nature of the programs themselves [15]. Data are also challenging to obtain due to inconsistency in state laws related to reporting; moreover, federal datasets do not report on disability or illness and its impact on children's learning and school attendance (e.g., National Center for Educational Statistics or National Institute of Health) [10]. The limited research that does exist describes H/H programs as frequently substandard, characterized by instructors who may lack the qualifications to teach critical core subject matters, and teachers with limited instructional time [10, 14, 16].

Kindergarten to 12th grade (K-12) online education (also frequently referred to as virtual or online K-12 schooling), constitutes an internet-facilitated means for children to maintain or further their education. Originating in the United States in 1995, online schooling now exists in all 50 states in a variety of different models (public, private, and charter). In 2019, more than 300,000 U.S. K-12 students received all their instruction online, with another 4.7 million participating in hybrid instruction where students engage in online learning in addition to traditional school-based instruction [17, 18].

Given the flexibility offered by online education, multiple researchers have hypothesized that it may provide affordances for students who qualify for H/H status and other critical populations [19–21]. High-quality online schooling can effectively establish a sense of community and promote differentiated instruction with personalized learning, qualities that may be lacking in traditional H/H services [22]. Several research studies have explored online schooling for children with disabilities, all of which describe poorer outcomes when compared to children without disability [19, 23–25]. However, these studies often rely on parent-reported data, small sample-sizes, and a range of outcome variables (i.e., course level grade to high school graduation) that limit their generalizability. Finally, to date, published research has mainly focused on the broad and diverse population of students who receive special education accommodations rather than their outcomes.

The goal of this study was to respond to this dearth of existing research [26, 27]. The present study uses a robust dataset to analyze how students with H/H designation performed in K-12 online classes, particularly as compared to non-H/H counterparts. We analyzed six years of academic data from one large, online K-12 school in the United States, identifying those middle and high school students (typically between the ages of 11 and 19) with H/H designation and those without such designation. Our analysis was primarily descriptive, with some comparisons between groups. The remainder of the paper addresses the methods and results of the analyses, concluding with implications for practitioners and educators.

## Methods

This Institutional Review Board (UF IRB#201802520) exempt study explored de-identified data comprising all Florida Virtual School (FLVS) students who registered for at least one online course between August 1, 2012 and July 31, 2018. FLVS is a fully accredited, statewide public school district offering more than 190 courses to students from kindergarten through 12th grade (FLVS, n.d.) in Florida and globally. The state of Florida is one of at least five US states that require students to complete at least one online course prior to graduation; FLVS is just one of several providers with which students may complete this requirement [28].

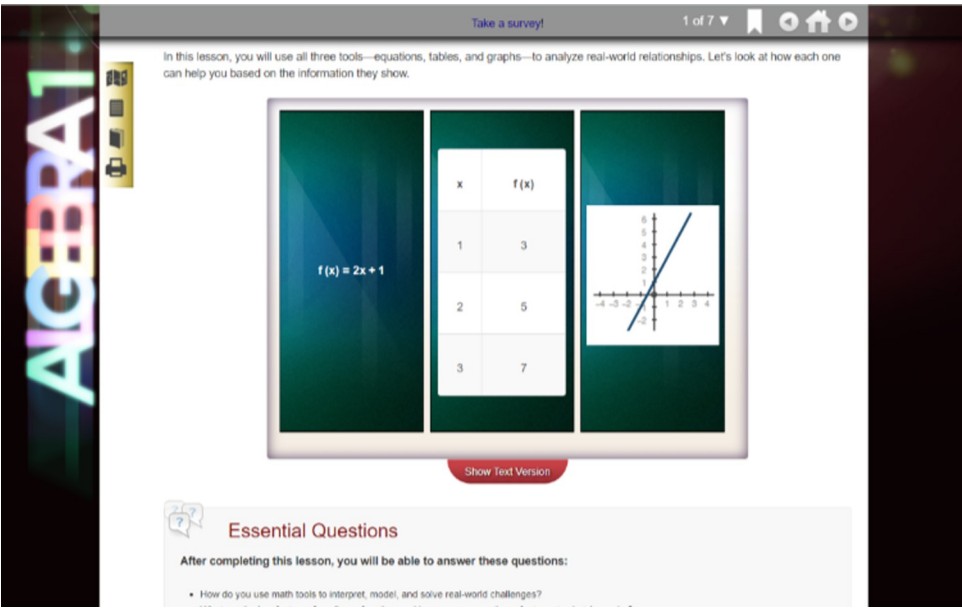

**Fig 1. FLVS Algebra 1, lesson overview.**

(Exemptions to this requirement are made for learners enrolled in special education programs with individual education plans that indicate online learning is inappropriate [29]).

As a not-for-profit organization, FLVS reinvests revenues into the development of new educational technologies and course development. The FLVS curriculum team designs state and national standards-aligned courses, incorporating best practices in educational technology and online learning [30]. Certified teachers draw on these courses to offer both synchronous (live) and asynchronous instruction (e.g., pre-recorded lessons) as well as individualized discussion-based assessments and support.

FLVS students have a variety of flexible and full-time enrollment options and can register for one or multiple courses. These courses include the core academic courses, electives, career and technical education courses, and Advanced Placement (college-level curricula) courses. Course content for middle and high school level courses is designed to cover 18 weeks of week (aligning with a standard semester). Importantly, while students may progress at their own pace through a course, students enrolled in 'core' courses (Algebra 1, Geometry, Biology, US History, Middle School Civics) must complete a statewide, standardized end-of-course examination. These examinations are offered over two-week periods five different times through a calendar year. FLVS designs its courses to interactive and incorporate games, videos, and multiple tools that aid learners in mastering course objectives [31]. Lessons begin with an overview of the content to be addressed (Fig 1 contains an example Algebra 1). Lessons also include activities and assignments to be completed individually or with peer feedback (see Figs 2 and 3 for examples from Algebra 2 and English 3).

Students have round-the-clock access to their courses allowing them to study anytime and anywhere. They master course objectives to complete the course and receive course credit Additionally, FLVS courses include accommodations, when and where applicable, such as extended time and as-needed opportunities for shortened assignments [32]. FLVS curricula, their online learning tools, and FLVS teachers have repeatedly received state, national and international accolades for online teaching and learning excellence [33].

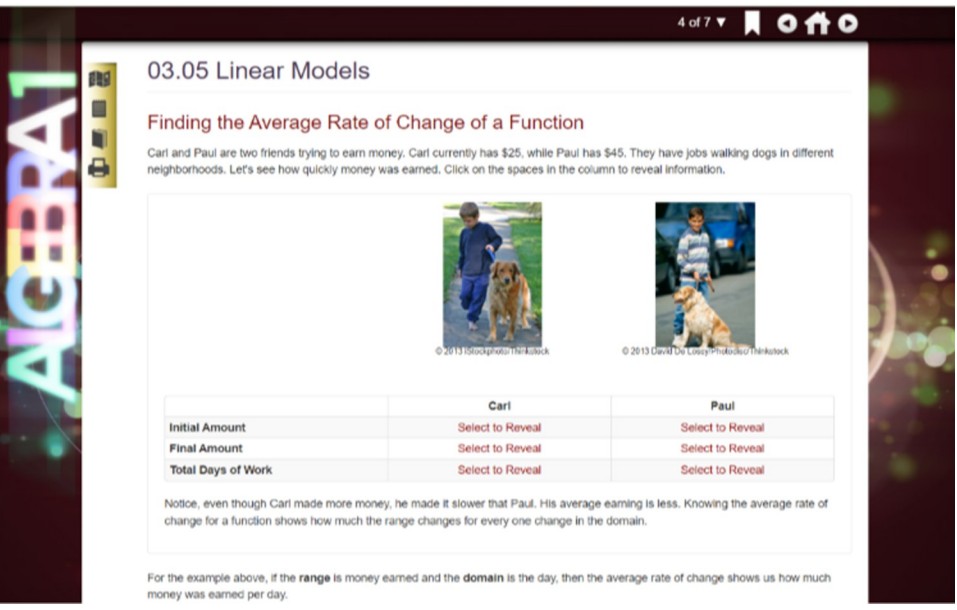

**Fig 2. FLVS Algebra 1, interactive guided practice.**

## Data and inclusion criteria

Prior to receiving the dataset, identifiers include name, address, email, phone number were removed by FLVS staff unassociated with the study. Given the study's IRB designation at exempt, the requirement for consent/assent was waived. The resultant dataset consisted of course instance data from two cohorts. Cohort one was comprised of all FLVS students with H/H status

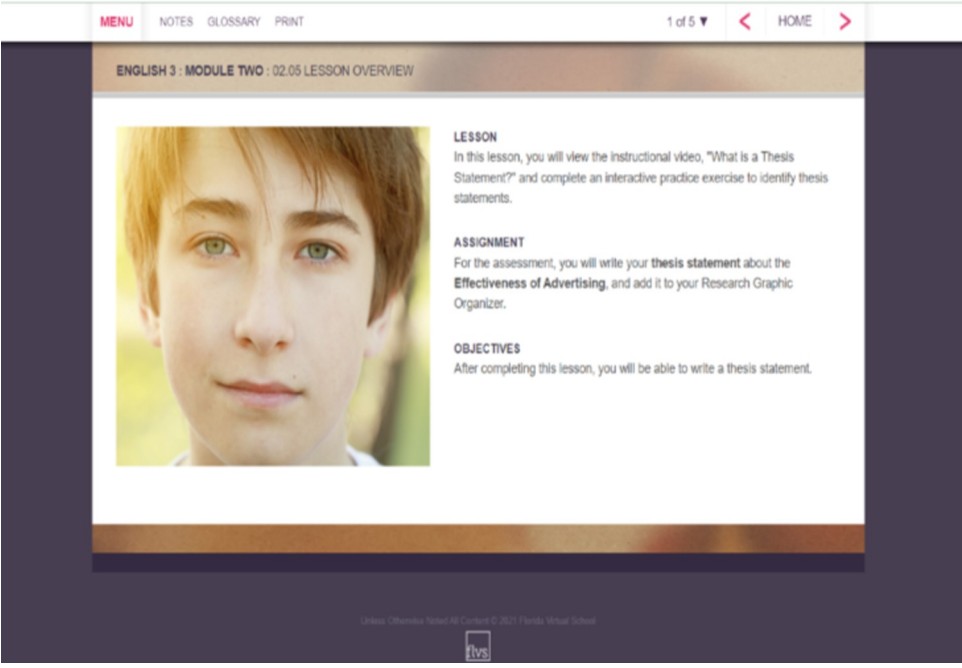

**Fig 3. Screenshot, FLVS English 3, lesson overview.**

from August 1, 2012 to July 31, 2018. Cohort two contained all non-H/H students enrolled during that same time frame. Non-H/H students included those individuals who were either full or part-time enrolled in one or more FLVS courses (see Table 1 for a listing of courses included in this dataset). This group included students who were designated as homeschooled students at

**Table 1. FLVS courses included in dataset.**

| |
|---|
| Adv Alg. w/ Financial Applications |
| AP Art History |
| AP Biology |
| AP Calculus AB |
| AP Calculus BC |
| AP Computer Science A |
| AP English Lit. and Composition |
| AP Environmental Science |
| AP Human Geography |
| AP Macroeconomics |
| AP Microeconomics |
| AP Psychology |
| AP Statistics |
| AP US Government and Politics |
| AP United States History |
| Algebra 1 |
| Algebra 2 |
| Algebra 2 (CR) |
| Anatomy and Physiology |
| Art History 1 Honors |
| Art in World Cultures |
| Biology 1 |
| Calculus Honors |
| Career Research/Decision Making |
| Careers in Fashion/Interior Design |
| Chemistry 1 |
| Chemistry 1 (CR) |
| Chinese 1 |
| Chinese 2 |
| Chinese 3 Honors |
| Computing for College and Careers |
| Creative Photography 1 |
| Criminal Justice Operations 1/Level 2 |
| Critical Thinking/Study Skills |
| Information Technology/Level 2 |
| Driver Education/Traffic Safety |
| Earth/Space Science |
| Economics with Financial Literacy |
| Econ. w/ Financial Literacy (CR) |
| English 1 |
| English 1 (CR) |
| English 2 |
| English 2 (CR) |

(*Continued*)

**Table 1.** (Continued)

| |
|---|
| English 3 |
| English 3 (CR) |
| English 4 |
| English 4 (CR) |
| English 4: Florida College Prep |
| Fitness Lifestyle Design |
| Forensic Science 1 |
| Foundations of Web Design/Level 3 |
| French 1 |
| French 2 |
| Geometry |
| Guitar 1 |
| Health 1 –Life Management Skills |
| HOPE–Physical Education |
| Intensive Reading |
| Intro. To Educational Technology |
| Journalism 1 |
| Latin 1 |
| Latin 2 |
| Law Studies |
| Leadership Skills Development |
| Liberal Arts Mathematics 1 |
| Liberal Arts Mathematics 2 |
| M/J Business Keyboarding |
| M/J Career Research/Decision |
| M/J Civics |
| M/J Comprehensive Science– 6/7 |
| M/J Comprehensive Science– 7/8 |
| M/J Comprehensive Science 1 |
| M/J Comprehensive Science 2 |
| M/J Comprehensive Science 3 |
| M/J Creative Photography 1 |
| M/J Critical Thinking & Learning |
| M/J Fitness–Grade 6 |
| M/J Grade 6 Mathematics |
| M/J Grade 7 Mathematics |
| M/J Grade 8 Pre-Algebra |
| M/J Guitar 1 |
| M/J Language Arts 1 |
| M/J Language Arts 2 |
| M/J Language Arts 3 |
| M/J Pre-Algebra |
| M/J Reading I |
| M/J Spanish, Beginning |
| M/J Spanish, Intermediate |
| M/J United States History |
| M/J World History |
| Marine Science 1 |

(*Continued*)

**Table 1.** (Continued)

| |
|---|
| Mathematics for College Readiness |
| Music of the World |
| Outdoor Education |
| Parenting Skills/Level 2 |
| Peer Counseling 1 |
| Peer Counseling 2 |
| Personal & Family Finance/Level 2 |
| Personal Fitness |
| Physical Science |
| Physics 1 |
| Pre-Calculus Honors |
| Psychology 1 |
| Reading for College Success |
| Social Media 1 |
| Sociology |
| Spanish 1 |
| Spanish 2 |
| Spanish 3 Honors |
| Spanish 4 Honors |
| Spanish 3 |
| Theatre & Film Production |
| United States Government |
| United States Government (CR) |
| United States History |
| User Interface Design |
| User Interface Design/Level 3 |
| World History |
| World History (CR) |

AP: Advanced Placement, CR: Credit Recovery, M/J: Middle/Junior High School Level.

the time of their enrollment. FLVS provided deidentified aggregate course outcomes data for both cohorts for the timeframe in question. To protect student privacy, FLVS does not collect student health information (i.e. medical condition(s) necessitating H/H designation) only those students who had received H/H designation by a designated healthcare professional and provided appropriate documentation were able to receive H/H designation at FLVS.

## Data analysis

Given the paucity of quantitative information about children with special healthcare needs and K-12 online learning, this study focused on two key areas. First, we set out to explore enrollment information of H/H students enrolled in FLVS courses. This included the number of students enrolled, the courses that they enrolled in, and their performance in these courses. After determining the size of the H/H cohort (n = 375), we limited the exploration of demographic data to avoid indirect student identification (in several cases only three or fewer H/H students were associated with a particular Florida county, age or age group in a specific year) and adhere with Federal Law (Family Educational Rights and Privacy Act, 20 U.S.C. § 1232g; 34 CFR Part 99). The second area of study was to examine H/H performance against traditional

online students enrolled during the same time frame and in the same types of courses as H/H students.

Prior to analysis, the data were reviewed for outliers, resulting in one H/H student's elimination from the dataset (this student was associated with more than 300 course instances in one semester, all of which resulted in withdrawals). During our analysis we defined characteristics of the data using specific terminology described by Cavanaugh et al. and Black et al. [34, 35], commonly used in the study of K-12 online learning, including:

1. *Aggregate Course Outcomes*: course name, year and semester/segment, and the frequency of grade and non-grade outcomes

2. *Course Completion*: a course instance where an A-F grade was generated;

3. *Course Instance*: a single course registration and associated de-identified information (course name, year and semester/segment of intended enrollment, and outcome);

4. *Course Non-Completion*: a course instance with a non-grade outcome (e.g. withdrawing from a course).

5. *Course Persistence*: a proportion of Course Completion to Course Non-Completion

Chi-square tests were used to explore dichotomous student outcomes (grade generating versus non-grade generating) by dichotomous status (H/H versus non-H/H status) drawn from the entire FLVS population. That is, we compared whether the categorical variables were independent of each other. Z-tests of proportion were explored to test whether two populations or groups differed significantly [36]. During instances where multiple statistics were run on the same dataset, we employed the Benjamini and Hochberg procedure to control for familywise Type I error [37]. We used descriptive statistics to describe non-categorical data. Data was analyzed using R and RStudio, and the tidyverse, and psych packages [38–40].

## Results

A total of 375 unique students (44.0% female, 56.0% male) were enrolled in the FLVS hospital-homebound (H/H) program between August, 2012 and July, 2018. The comparison group, comprised of non-H/H students enrolled in one or more FLVS courses during the same time period totaled 1,191,508 unique students (57.0% female, 43% male). We did not explore race and ethnic data from the H/H cohort, but in 2019–2020 FLVS reported that 70.2% of part-time enrollees identified as White, 16.1% as Black of African American, 6% as Multi-Racial, and 4.5% as Asian. The H/H cohort was comprised of learners from 25 different Florida counties, two university-associated lab schools, and homeschooled students. Whereas the comparison group represented students from all 67 Florida school districts. Among both cohorts the majority of learners lived in more populous counties (e.g. Miami-Dade, Broward, Orange, Hillsborough).

There were a total of 2,534 H/H course enrollments; each student was associated with a mean of 5.8 course instances (SD = 6.8, median = 3, mode = 2). Among these 375 students, 26.6% (n = 100) had only one semester of enrollment or attempted enrollment in an FLVS course. Non-H/H students were enrolled in an average of 4.6 courses (SD = 5.8. median = 3, mode = 1).

### Finding #1: Students designated as H/H performed similarly to non-H/H designated counterparts across core content areas

Courses in which A-F grades were generated (n = 922 for H/H; n = 2,381,607 for non-H/H) were evaluated to compare student outcomes across all course enrollment. Overall, students

**Table 2. Graded performance by course instance (H/H vs Non-H/H designated students, 2012–2018).**

| H/H Students | | | Non-H/H Students | | | | | |
|---|---|---|---|---|---|---|---|---|
| Outcome | Frequency | Percent | Outcome | Frequency | Percent | Z | p | adj p (Benjimini and Hochberg)** |
| A | 481 | 52.17% | A | 1,284,619 | 53.94% | 1.16 | .28 | .28 |
| B | 266 | 28.85% | B | 733,520 | 30.80% | 1.64 | .19 | .28 |
| C | 125 | 13.56% | C | 279,801 | 11.75% | 2.90 | .08 | .14 |
| D | 31 | 3.36% | D | 52,899 | 2.22% | 5.52 | .01 | .05 |
| F | 19 | 2.06% | F | 30,768 | 1.29% | 4.27 | .03 | .08 |

** Critical p values adjusted to address family-wise error.s

designated as H/H received grades that did not significantly differ from non-H/H students. In other words, while non-significance cannot be interpreted as support for no difference between groups, H/H students in this sample performed similarly, even given challenges associated with chronic illness or disabilities, see Table 2.

This finding needs to be contextualized within a broader understanding of the types and rigor of courses being taken by H/H students. Table 3 presents course completion data for the 10 most frequent course registrations by H/H students as well as comparative data for non-H/H students. Two outcomes are apparent. First, H/H students enrolled in similar core courses as their non-H/H counterparts. Standard diploma requirements in Florida include English/Language Arts, Math, Science, Social Studies (including government and economics), Physical Education (e.g., HOPE in the table below), and Fine Arts. H/H students demonstrate similar levels of course completion compared to non-H/H students in most core topics. The second evident result from this table relates to course selection and functionality of H/H students. In addition to core courses, H/H students take and complete life-skills courses like Drivers Education in relatively the same manner as their non-H/H peers. In sum, H/H students took the same types and rigor of courses and still performed similarly to their non-H/H counterparts in this study.

**Table 3. Performance by course instance for the 10 most frequent course registrations by H/H students (with comparison to Non-H/H designated students) 2012–2018.**

| Course | H/H Instances | H/H Completion | H/H Non-Completion | Proportion Complete | Non-H/H Instances | Non-H/H Completion | Non-H/H Non-Completion | Proportion Complete | P[a] | adj p (Benjimini and Hochberg)[b] |
|---|---|---|---|---|---|---|---|---|---|---|
| Driver Education/Traffic Safety–Classroom | 145 | 80 | 65 | 55% | 459,070 | 280,672 | 178,398 | 61% | 0.142 | NA |
| Algebra 1 | 108 | 39 | 69 | 36% | 191,866 | 49,916 | 141,950 | 26% | 0.016 | 0.14 |
| English 1 | 105 | 21 | 84 | 20% | 150,789 | 39,894 | 110,895 | 26% | 0.134 | NA |
| Biology 1 | 99 | 21 | 78 | 21% | 135,004 | 35,488 | 99,516 | 26% | 0.25 | NA |
| Spanish 1 | 95 | 33 | 62 | 35% | 278,820 | 122,388 | 156,432 | 44% | 0.072 | NA |
| World History | 92 | 25 | 67 | 27% | 175,048 | 56,821 | 118,227 | 32% | 0.27 | NA |
| Economics with Financial Literacy | 86 | 47 | 39 | 55% | 67,722 | 33,926 | 33,796 | 50% | 0.39 | NA |
| Algebra 2 | 81 | 29 | 52 | 36% | 177,607 | 61,948 | 115,659 | 35% | 0.86 | NA |
| United States Government | 80 | 22 | 58 | 28% | 77,858 | 35,205 | 42,653 | 45% | 0.002 | 0.02 |
| HOPE—Physical Education (Core) | 78 | 49 | 29 | 63% | 326,285 | 181,749 | 144,536 | 56% | 0.2 | NA |

a: All comparisons were conducted using a Z-test of proportion.

b: Critical p values adjusted to address family-wise error.

**Table 4. Non-grade outcomes by course instance (H/H v non-H/H designated enrollments, 2012–2018).**

| H/H Designated Course Instances | | | Non-H/H Course Instances | | | | | |
|---|---|---|---|---|---|---|---|---|
| Outcome | Frequency | Percent | Outcome | Frequency | Percent | \|z\| | p | adj p (Benjimini and Hochberg)[a] |
| NA[b] | 578 | 35.86% | NA | 1,635,820 | 53.0% | 13.75 | < .001 | < .001 |
| NAc[c] | 3 | 0.19% | NAc | 10,760 | 0.3% | 1.1 | 0.27 | 0.27 |
| W[d] | 472 | 29.28% | W | 363,249 | 11.8% | 21.84 | < .001 | < .001 |
| WNG[e] | 50 | 3.10% | WNG | 177,898 | 5.8% | 4.58 | < .001 | < .001 |
| WF[f] | 131 | 8.13% | WF | 133,810 | 4.3% | 7.5 | < .001 | < .001 |
| WP[g] | 248 | 15.38% | WP | 416,121 | 13.5% | 2.25 | 0.001 | 0.048 |
| NULL[h] | 130 | 8.06% | NULL | 351,326 | 11.4% | 4.18 | < .001 | < .001 |

a: Critical p values adjusted to address family-wise error.

b: NA: Student had verification from guidance but was never assigned to a teacher. This status occurs when student does not reply to course request message. FLVS subsequently removes the course request. Student will need to re-register.

c: NAc: Student was assigned an instructor, but was removed from course (may be initiated by teacher or student). Please note: This often occurs when students do not respond to welcome call attempts by their teacher.

d: Withdrawn (W/WNG): Student has been withdrawn during the 14-day drop/add period with no penalty. No credit is awarded.

e: WF: Student withdrew from course with a grade of less than 60% after the 14-day drop/add period expired. No credit is awarded.

f: WP: Student withdrew from course with a 60% or higher grade after the 14-day drop/add period expired. No credit is awarded.

g: NULL: No data provided.

## Finding #2: Student course enrollments resulting in a grade were significantly different between H/H students and non-H/H students

Of the 2,534 H/H course enrollments, 36.4% (n = 922) resulted in a final grade ranging between "A" and "F" (Table 4). Comparatively, 43.5% of non-H/H student enrollments (n = 2,281,607) resulted in a grade. When analyzed, these data suggest that courses taken by H/H designated students were 25.9% (95% CI: 2.5–97.5%) less likely to result in grade generation as compared to courses taken by non-H/H students (p < .01; Table 4).

Student completion rates in online courses may be low for a variety of reasons [41]. For H/H designated students, course instances resulting in non-graded outcomes were most frequently due to failure on the part of the student or family to complete their enrollment (despite having authorization from the school of record; 35.9%) or withdrawal from the course during the 14-day drop/add period (29.3%). A complete list of non-grade outcomes by course instance between H/H and non-H/H students is presented in Table 5.

## Finding #3: H/H student completion rates were positively correlated with two important aspects of enrollment

On average, course completion rate for H/H students was 0.36 for *all* courses. That number climbs to 0.56 when examining courses where five or more students were enrolled suggesting that the more courses taken by the H/H student translates to a higher likelihood of completion. This translates to a statistically significant, positive correlation between student enrollments

**Table 5. H/H vs. non-H/H designated outcomes by course instance (grade generation vs. non-grade generation).**

| H/H Enrollments/Instances | | Non-H/H Enrollments/Instances | | |
|---|---|---|---|---|
| n (%) | | n (%) | | P |
| Grade | 922 (36.4%) | Grade | 2,381,607 (43.53%) | <0.01 |
| Non-Grade | 1,612 (64.00%) | Non-Grade | 3,088,984 (56.47%) | |

and completion ratios (r = .79, p < .01) suggesting that the presence of peers in an online class-room is associated a higher likelihood of completion. Similar analyses were conducted on course completion by student. These data provide evidence that, on average, the average rate of course completion on a per H/H student is .49 (sd = .37), but when reviewing the completion ration of H/H students with five or more course instances (n = 140), that number climbs to .62 (sd = .29). This translates to a statistically significant, positive correlation between student enrollments and completion ratios (r = .75, p < .01), suggesting that the more courses taken by the H/H student translates to a higher likelihood of completion.

## Discussion

K-12 online schooling continues to expand, evolve, and become a mainstay in education around the world. It saw significant growth during the COVID-19 pandemic, which impacted parent, teacher, and student perspectives of online learning. For instance, COVID has encouraged many parents to consider alternatives to traditional means of education, particularly the care-givers of students who may have health concerns such as those with respiratory disease or who are immunocompromised [42–46].

However, even with the increased recent focus, since the inception of K-12 online education, there has been little attention paid to supporting the needs of students with chronic illnesses or disabilities [47]. There are few studies of students who have been afforded school accommodations specific to their healthcare needs [24]. This is due to myriad barriers to research including different data systems, different definitions of health care needs compared to educational needs, and a lack of grade standardization across educational institutions. Specific to virtual schools, prior research related to students with disabilities frequently relied on convenience sampling, qualitative data, or parents' willingness and ability to communicate their child's health status and learning outcomes [19]. The studies that do exist provide evidence that learners with special healthcare needs comprise a significant subset of the population of students attending K-12 online schools in the United States [25].

The present study used a rigorous inclusion criterion (hospital/homebound status) to identify an extremely vulnerable, high risk population. These criteria allowed the research team to reduce concerns for self/parent-selection bias, convenience sampling, and survivorship bias. In doing so, this study represents the first, systematic study of a large virtual schooling population that compares traditional (non-H/H designation) students to those with legally documented health care needs.

Each of the three outcomes of this study lends important insight into the study of H/H students taking online courses for their educational continuity. First, H/H students were as successful in online schooling as their non-H/H counterparts in this study. This included both functional (e.g., driver's education) classes and nearly all core courses (e.g., Algebra and Economics). These results support prior, descriptive, research that provided evidence that FLVS H/H students achieved similar grade associated outcomes [19, 23–25, 48]. This means that practitioners and parents should consider K-12 online schooling as a valid opportunity for students who are unable to attend in person.

It is worth noting that not every H/H student will be successful, just like not every non-H/H student is successful in either face-to-face or online schooling. In this study, H/H students had a significantly different course-to-completion percentage (36.4%) than their non-H/H counterparts (43.53%). However, H/H students taking online classes in this study were more successful when taking multiple courses. Parents, practitioners, and advisors should consider this when enrolling H/H students in online courses.

## Broader implications and future research

H/H students accounted for less than 0.1% of total FLVS course instances during the six years associated with this analysis. The reality is that H/H students represent a population under-served by many education systems, including online education. The low enrollments here may point to a lack of awareness of the opportunity that online schooling can provide. Data find-ings suggest that practicing pediatric healthcare professionals should be made aware of the positive potential outcomes for their patients.

There are several areas for future and related research. First, the data analyses provided evi-dence that H/H students who participated in five or more courses were more likely to com-plete a course. Similarly, those courses that had more than five H/H student enrollments during the study period were also more likely to result in H/H student completions. It is not enough to determine this happened; researchers need to further explore why this happened. Such outcomes would then impact policy recommendations for course enrollment of H/H stu-dents (e.g., many states have a requirement that all students participate in at least one online course prior to high school graduation).

A second area for future investigation relates to differences in registrations, enrollments, and de-enrollments of courses. H/H designated students were significantly more likely to with-draw from a course than non-H/H designated students (p < .01). This was true regardless of whether the student withdrew before or after the 14-day drop/add period and irrespective of whether they had a passing or failing grade at the time of withdrawal. The data presented in this study are not able to explain this discrepancy. Possible causes include improvement in the students' health allowing for a return to their traditional schooling environment, academic challenges unrelated to health, or a decline in student health. Future research should include interviews of H/H students—both those who have completed courses and those who were unable to complete courses—to better understand learner experiences.

Conversely, among non-H/H designated students, 53% of course instances without grades are associated with incomplete registration processes; that is, the student had verification from their guidance office, but was never assigned to a teacher. This status occurs when the student or parent does not reply to course request message. The online school (in this case, FLVS) sub-sequently removes the course request. It is noteworthy that the rate associated with incomplete registrations was significantly lower within the H/H population (p < .001), which, again, reflects an area for future investigation. Recent research provides evidence that the odds of stu-dent disengagement and subsequent non-completion in an online learning environment drop by 50% during first 25–50 minutes of use [49]. Future research could examine specific prac-tices to support H/H students during this timeframe.

A final area for subsequent research relates to pediatric investigations of H/H online stu-dents in terms of both disease/disability management and overall mental health. Child health professionals need to understand the educational outcomes of their patients as they do health outcomes; and, at a minimum, they need to understand intended goals for their patients when they document their medical necessity. Certainly, educational outcomes will be less prioritized than medical outcomes and procedures that are lifesaving, yet child health providers cannot lose sight of this important goal at a time when children are surviving (and surviving well) many of the conditions they might not have survived even a decade or two earlier.

Having acknowledged these educational interests, literature surrounding medical outcomes and clinical trials may not necessarily overlap with online schooling research. Yet there are promising opportunities for increased collaboration. For instance, clinical research has pro-vided evidence that innovative technologies like virtual reality can significantly support pain management, and a further outcome could be educational attainment (e.g., Malloy & Milling)

[50]. Public health research has also provided evidence that social network communities (like those presented in online learning management systems) can provide significant support for mental health, a common reason for choosing virtual schooling [51]. Yet there is scarce inter-professional research to date that takes the next step and examines the impact of virtual schooling and online education for adolescent mental health and positive medical outcomes like pain or disease management.

## Limitations

It is important to acknowledge limitations associated with this study. This study explored data from one virtual school, thus limiting generalizability. FLVS does not collect private health information about its students; therefore, the analysis presented here did not include information about morbidities or acuity. Analyses also excluded demographic information, which could have resulted in inadvertent identification. It also did not include behavioral variables related to student use of online learning materials (i.e., time on task and task repetition). These exclusions limited our understanding of the H/H population. Finally, our comparison population may have included children living with disabilities and special healthcare needs who do not qualify or have not received hospital/homebound designation.

To further address these limitations, future research should include a more granular analysis of students, student wellness, and learning outcomes. Future research should also incorporate methods such as survival analysis, commonly employed in sociology and epidemiology, but increasingly found in educational studies to explore and better predict learning attrition and the efficacy of interventions to address it [44, 52, 53]. Importantly, there is likely a selection bias associated with students, both those qualified for H/H and 'typical' students who enroll in K-12 online schools, resulting in less racial and ethnic diversity and more affluence than students in traditional learning environments. Finally, it is critical to note that this analysis did not incorporate the varied nature of course curricula as an independent variable. Curricula, even same subject matter curricula, can vary significantly over time and between instructors.

## Implications for policy

The H/H student population represents a vulnerable, underserviced, and understudied population and a unique opportunity for research into the intersection of health and education. Fundamental questions about the impact of learning on health and quality of life outcomes in the short and long-term, appropriate teacher training, educating healthcare professionals, best practices for delivering homebound services, and H/H student performance and engagement in online and non-online environments need further exploration_msocom_1 [15, 25, 54]. Additionally, we have little understanding of the affordances and challenges associated with learning in an online environment for H/H students, which likely vary considerably based upon the unique needs of the child. Nonetheless, this study is one of the first projects to bridge these streams of literature and illustrate that H/H students have educational success with online schooling. Healthcare providers may find that online school can serve as a permanent intervention that comes alongside students in ways other interventions have fallen short.

## Author Contributions

**Conceptualization:** Erik W. Black, Richard E. Ferdig, April Fleetwood, Lindsay A. Thompson.

**Formal analysis:** Erik W. Black, Richard E. Ferdig, April Fleetwood, Lindsay A. Thompson.

**Investigation:** Erik W. Black.

**Methodology:** Erik W. Black, Richard E. Ferdig, April Fleetwood, Lindsay A. Thompson.

**Project administration:** Erik W. Black, April Fleetwood.

**Resources:** April Fleetwood.

**Writing – original draft:** Erik W. Black, Richard E. Ferdig, April Fleetwood, Lindsay A. Thompson.

**Writing – review & editing:** Erik W. Black, Richard E. Ferdig, April Fleetwood, Lindsay A. Thompson.

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
