## [Decision Letter · Decision Letter 0]

10 Dec 2021

PONE-D-21-36433Hospital Homebound Students and K-12 Online SchoolingPLOS ONE

Dear Dr. Black,

Thank you for submitting your manuscript to PLOS ONE. After careful consideration, we feel that it has merit but does not fully meet PLOS ONE’s publication criteria as it currently stands. Therefore, we invite you to submit a revised version of the manuscript that addresses the points raised during the review process.

We look forward to receiving your revised manuscript.

Kind regards,

Rong Zhu, Ph.D.

Academic Editor

PLOS ONE

Journal Requirements:

2. In ethics statement in the manuscript and in the online submission form, please provide additional information about the records used in your retrospective study. Specifically, please ensure that you have discussed whether all data were fully anonymized before you accessed them and/or whether the IRB or ethics committee waived the requirement for informed consent. If patients provided informed written consent to have data from their medical or educational records used in research, please include this information.

"April Fleetwood is employed by Florida Virtual School." 

Reviewers' comments:

Reviewer's Responses to Questions

**Comments to the Author**

1. Is the manuscript technically sound, and do the data support the conclusions?

Reviewer #1: Partly

2. Has the statistical analysis been performed appropriately and rigorously? 

Reviewer #1: I Don't Know

3. Have the authors made all data underlying the findings in their manuscript fully available?

Reviewer #1: Yes

4. Is the manuscript presented in an intelligible fashion and written in standard English?

Reviewer #1: Yes

5. Review Comments to the Author

Reviewer #1: This study compared the performance of hospital homebound students with regular students enrolled in an online learning school. The topic of this study is definitely worth investigating, as very little evidence exists on the academic performance of hospital homebound students in education – especially in online educational courses – compared against a cohort of regular students. That being said, the topic of this study is of relevance to a broad international field of researchers and may contribute significant findings to the field. However, the current version of the manuscript lacks several important aspects such as (a) a documentation of the data analysis in the methods section, (b) a clear inclusion criteria section of how the samples were drawn and (d) a description of the online school. Not every reader will be aware of the online school and thus, it should be described in more detail (a separate section). Based on these key aspects, it is difficult to interpret the findings currently. Finally, the current version of the manuscript misses some important contributions regarding online learning systems which may be cited in a revised version of the manuscript.

Introduction:

The introduction is well-written, and I have only a few comments here.

Typo page 2:

The sentence: “Meeting the educational needs…” appears to miss some words at the end. I recognized some more typos while reading the manuscript. The authors should proof read the manuscript.

The author may want to provide an overview or “the present study” section at the end of the introduction, which helps the reader to grasp an overview of what this study is about.

Methods:

Online schools and Online Courses:

The online school should be described in more detail. No information at all in provided on which courses are offered, for which age/grade level, which subjects, etc. It should also be noted whether students receive feedback or how students get learning materials assigned. At the current version of the manuscript, no information at all is provided on how an “online course” looks like or what is meant by and “online course”.

Inclusion criteria:

A separate inclusion criteria section which describes the inclusion process may help the reader to grasp the inclusion process of the two cohorts better.

Data Analysis:

The data analysis and the motivation behind the data analysis is not described at all. Currently, the manuscript lacks any description of why the analyzes were conducted. Why did the authors examine these analyses and not others? Or were other analyzes performed but did not reach significance and were thus not reported? The motivation of why the dependent/independent variables were selected should be noted.

The statistical software the analyzes were carried out with should be noted.

Results:

The authors should provide more descriptive variables on the data. What was the mean number of courses students enrolled in (range of enrolled courses)? How long does one course take? How many sessions does one course include (range)? How many different courses exist in this online school? Were all courses analyzed? More information here is needed.

Online courses in general vary heavily and information on this is necessary to compare this study with others describing performance outcomes in online courses.

Non-significant findings cannot be interpreted as support for no differences between groups! The authors should note this and may want to carry out a Bayesian Analysis to support their point or compute a Bayes Factor for their findings.

Discussion:

The authors analyzed whether students remained enrolled or not. They may want to hint towards recent attempts addressing this by means of survival analyzes. The lack of a systematic and persistent usage of online courses is one of the main problems of online courses and attempts on how to increase the enrollment for longer periods of time is one of the most challenging future questions in the field of online courses (Bacca-Acosta & Avila-Garzon, 2021; Grutzmacher et al., 2019; Plank et al., 2008; Spitzer et al., 2021; Villano et al., 2018).

The authors note the boom in online learning courses around the world due to the COVID-19 pandemic. They may want to bolster this argument by adding some more references to it providing evidence that several online learning environments experienced a usage boom since march 2020 (Meeter, 2021; Spitzer et al., 2021; Tomasik et al., 2020; Velde et al., 2021).

The authors may want to note that they did not control for any behavioral variables in their analysis such as time on task, number of problem sets, or repetition attempts students may have used. Future research may investigate differences in usage patterns between the two cohorts.

References

Bacca-Acosta, J., & Avila-Garzon, C. (2021). Student engagement with mobile-based assessment systems: A survival analysis. Journal of Computer Assisted Learning, 37(1), 158–171. https://doi.org/10.1111/jcal.12475

Grutzmacher, S. K., Munger, A. L., Speirs, K. E., Vafai, Y., Hilberg, E., Duru, E. B., Worthington, L., & Lachenmayr, L. (2019). Predicting attrition in a text-based nutrition education program: Survival analysis of Text2bhealthy. JMIR MHealth and UHealth, 7(1), 1–10. https://doi.org/10.2196/mhealth.9967

Meeter, M. (2021). Primary school mathematics during Covid-19: No evidence of learning gaps in adaptive practicing results. Trends in Neuroscience and Education, 25(October), 100163. https://doi.org/10.1016/j.tine.2021.100163

Plank, S. B., DeLuca, S., & Estacion, A. (2008). High school dropout and the role of career and technical education: A survival analysis of surviving high school. Sociology of Education, 81(4), 345–370. https://doi.org/10.1177/003804070808100402

Spitzer, M. W. H., Gutsfeld, R., Wirzberger, M., & Moeller, K. (2021). Evaluating students ’ engagement with an online learning environment during and after COVID-19 related school closures : A survival analysis approach. Trends in Neuroscience and Education, 25, 100168.

Tomasik, M. J., Helbling, L. A., & Moser, U. (2020). Educational gains of in-person vs. distance learning in primary and secondary schools: A natural experiment during the COVID-19 pandemic school closures in Switzerland. International Journal of Psychology, 56(4), 566–576. https://doi.org/10.1002/ijop.12728

Velde, M. van der, Sense, F., Spijkers, R., Meeter, M., & ... (2021). Lockdown Learning: Changes in Online Study Activity and Performance of Dutch Secondary School Students during the COVID-19 Pandemic. https://psyarxiv.com/fr2v8/

Villano, R., Harrison, S., Lynch, G., & Chen, G. (2018). Linking early alert systems and student retention: a survival analysis approach. Higher Education, 76(5), 903–920. https://doi.org/10.1007/s10734-018-0249-y

6. PLOS authors have the option to publish the peer review history of their article (what does this mean?). If published, this will include your full peer review and any attached files.

Reviewer #1: No

---

## [Author Response · Author response to Decision Letter 0]

23 Jan 2022

To the Editor and Reviewers,

I am pleased to submit this revised manuscript, Hospital Homebound Students and K-12 Online Schooling, for review. I appreciate the insightful suggestions made by the editor and reviewers, we have worked to address each suggestion to the best of our ability. The authors received no specific funding for this work.

In response to a comment by the editor, we have included a more detailed explanation of data availability. Importantly, our ability to provide access to the data is limited by Florida law. Family Educational Rights and Privacy Act (FERPA), 20 U.S.C. § 1232g; 34 C.F.R Part 99; and chapter 119 and section 1002.22, Florida Statutes states: Individuals and organizations that receive data through the research request process have strict obligations to protect that data and must meet the requirements of both federal and state laws in their usage and handling of that data. The parties receiving this data have separate obligations based on these provisions and must protect the confidentiality and privacy of this information and may not disclose this information to others.

The Florida Department of Education requires all requests to comply with the Family Educational Rights and Privacy Act’s (FERPA’s) research exception (34 CFR Part 99.31[6][i]), which requires that the disclosure of information be limited to organizations conducting studies for, or on behalf of, educational agencies. In addition, the research must meet one of the following three allowable purposes: 1) to improve instruction; 2) to develop, validate or administer predictive tests; or 3) to administer student aid programs. Requests must meet one of these criteria to be approved.

In sum, the data is available via public request via Florida’s Government-in-the-Sunshine law. This provides the right of access to governmental documents at the state and local level. Data access can be facilitated using the Florida Department of Education External Research Data Request Process: https://www.fldoe.org/accountability/accountability-reporting/external-research-requests/. While this is not standard, we feel that this does not alter our adherence to PLOS ONE policies on sharing data and materials.

I have attached a table (see uploaded revision letter) that lists each suggestion, along with our response, inclusive of page numbers where the narrative addressing the response can be found.

---

## [Decision Letter · Decision Letter 1]

18 Feb 2022

Hospital Homebound Students and K-12 Online Schooling

PONE-D-21-36433R1

Dear Dr. Black,

We’re pleased to inform you that your manuscript has been judged scientifically suitable for publication and will be formally accepted for publication once it meets all outstanding technical requirements.

Kind regards,

Rong Zhu, Ph.D.

Academic Editor

PLOS ONE

Additional Editor Comments (optional):

Reviewers' comments:

Reviewer's Responses to Questions

**Comments to the Author**

1. If the authors have adequately addressed your comments raised in a previous round of review and you feel that this manuscript is now acceptable for publication, you may indicate that here to bypass the “Comments to the Author” section, enter your conflict of interest statement in the “Confidential to Editor” section, and submit your "Accept" recommendation.

Reviewer #1: All comments have been addressed

2. Is the manuscript technically sound, and do the data support the conclusions?

Reviewer #1: Yes

3. Has the statistical analysis been performed appropriately and rigorously? 

Reviewer #1: Yes

4. Have the authors made all data underlying the findings in their manuscript fully available?

Reviewer #1: No

5. Is the manuscript presented in an intelligible fashion and written in standard English?

Reviewer #1: Yes

6. Review Comments to the Author

Reviewer #1: The authors addressed all of my comments and I see no major flaws. Major parts of the manuscripts have been revised in great detail and all major points I previously raised have been addressed. Congrats!

7. PLOS authors have the option to publish the peer review history of their article (what does this mean?). If published, this will include your full peer review and any attached files.

Reviewer #1: No

---

## [Editor Report · Acceptance letter]

15 Mar 2022

PONE-D-21-36433R1 

Hospital Homebound Students and K-12 Online Schooling 

Dear Dr. Black:

I'm pleased to inform you that your manuscript has been deemed suitable for publication in PLOS ONE. Congratulations! Your manuscript is now with our production department. 

Kind regards, 

on behalf of

Dr. Rong Zhu 

Academic Editor

PLOS ONE